# Trifid Mandibular Condyle: Case Report and Current Review of the Literature

**DOI:** 10.3390/life12070976

**Published:** 2022-06-29

**Authors:** Nour Zoabi, Lazar Kats, Alon Ram, Alona Emodi-Perlman

**Affiliations:** 1Department of Oral Pathology, Oral Medicine and Maxillofacial Imaging, The Maurice and Gabriela Goldshleger School of Dental Medicine, Tel Aviv University, Tel Aviv 6139001, Israel; dr.nour.zoabi@gmail.com (N.Z.); lazarkat@tauex.tau.ac.il (L.K.); 2The Maurice and Gabriela Goldshleger School of Dental Medicine, Tel Aviv University, Tel Aviv 6139001, Israel; alonram@mail.tau.ac.il; 3Department of Oral Rehabilitation, The Maurice and Gabriela Goldshleger School of Dental Medicine, Tel Aviv University, Tel Aviv 6139001, Israel

**Keywords:** trifid mandibular condyle, bifid, trauma, cone beam CT imaging, temporomandibular joint (TMJ), temporomandibular disorders (TMD)

## Abstract

Trifid mandibular condyle (TMC) is a rare anatomical variation characterized by the duplication of the mandibular condyle. The aim of this study is to report a new case of a 26-year-old female patient with a left TMC and to review the current existing literature on TMC, the relevant cases, etiology, symptoms and different treatment modalities. The database engines PubMed, EMBASE, Scopus, Web of science, Scientific Electronic Library Online, Cochrane and CINAHL were searched for TMC cases from inception until April of 2022. Only 13 previous cases of TMC were found. Although it is a rare anatomical entity, TMC is increasingly being detected due to more advanced imaging techniques, especially computed tomography (CT), cone beam CT (CBCT) and magnetic resonance imaging (MRI) emerging in the field of dentistry. The etiology and pathogenesis of TMC and its relationship with TMD are still unclear. Further studies and follow-up may help to better understand this anatomic variant and possible interactions with local pathologies.

## 1. Introduction 

Trifid mandibular condyle (TMC) is a rare and uncommon anatomical variation [1,2,3,4,5,6,7,8] in which the head of the condyle is split into three distinct heads via grooves of variable depths. Most of the existing literature relates to the bifid variant (BMC) characterized by the duplication of the mandibular condyle [9,10]. According to the literature, the prevalence of BMC ranges from 0.018% to 1.82% [11,12,13] in the general population and may be unilateral with left side predominance or a less frequent bilateral variant. BMC was first reported on dried specimens by Hrdlicka back in 1941 [14]. The etiology and pathogenesis of this anatomical abnormality is debated and till today uncertain. According to the existing literature, there are two main possible etiologies, one of which is trauma, causing fractures of condylar regions, with the multiple heads being the result of healing and remodeling. The second possible etiology is thought to be developmental abnormality [15]. Other health conditions such as endocrine disorders, nutritional deficiency, infection, trauma, irradiation and genetic factors were found to be linked to TMC [16]. TMC is more rarely found than BMC and is usually an incidental inspected finding diagnosed during routine radiographic examinations and typically has no distinct clinical symptoms [1,2,3,4,17,18,19]. The most common signs and symptoms associated with BMC include hypomobility, arthralgia, intra articular noises and ankylosis [15], while the ones found to be associated with TMC include pain, anterior open bite and temporomandibular disorder signs such as: deviation of the mandible while opening, jaw weakness, mandibular hypoplasia, restricted mouth opening and clicking joint noises. Other reported signs include snoring during sleep, Frey’s syndrome, facial asymmetry, swelling of the masseteric region and incisal midline shift [1,2,3,4,5,6,7,8,16,17,20,21].

The aim of this study is twofold:(1)To report a new case of a 26-year-old female patient with a left TMC.(2)To review the existing literature on TMC, the relevant cases, etiology, symptoms and proposed treatments.

## 2. Case Report

A 26-year-old healthy female patient (Figure 1) was referred to the student’s orofacial pain clinic at the Tel-Aviv university dental school complaining about pain upon chewing on the right side of the mandible. The pain first appeared 3 years ago. It felt like a dull non-continuous pain which resolved spontaneously. The patient also complained of a clicking noise arising from the right jaw and a self-resolving open lock of the right temporomandibular joint (TMJ). The patient was asked whether she had any previous trauma or accident involving her jaw or face. The patient recalled an accident that occurred when she was 4 years old, she fell on her face from the top bunk of a bed; she recalls facial pain and restriction of jaw movements that lasted only for a few days and as far as she could remember it resolved without any special treatment.

Extra-oral examination demonstrated, facial asymmetry with deviation of the chin to the left (Figure 1 and Figure 2). 

Clinical examination was carried out according to the diagnostic criteria for temporomandibular disorders (DC/TMD) [22]. The examination revealed left uncorrected deviation of the mandible while opening. Passive opening—42 mm. Active opening—39 mm. Provocation of right familiar pain in the Masseter muscles occurred while opening the mouth and during palpation of the right inferior area of the Masseter muscle with referral to tooth no. #48. A non-familiar pain was provoked upon palpation of the lateral pole of the left temporomandibular joint (TMJ), left masseter, temporalis and right temporalis muscles. During opening, a non-painful crepitus was detected while palpating the left TMJ. Intra-oral examination demonstrated: Linea alba on the right and left buccal mucosa scalloped tongue, occlusion class 1 on the right side, class 2 malocclusion on the left side; 3 mm right midline shift (Figure 3). Fully buccal eruption of tooth no. #18 and mesial eruption of tooth no. #48. Pericoronitis signs were observed around tooth no. #48. The pain arising from the right masseter was thought to be caused either by referred pain from pericoronitis of tooth no. #48, and/or due to temporomandibular disorder (TMD), Myofacial pain with referral. Panoramic X-ray (Figure 4) demonstrated a partially impacted right and left lower wisdom tooth and a left bifid mandibular condyle with suspected lesion which seemed in accord with the clinical findings: uncorrected deviation to the left while opening, crepitus of left TMJ and pain upon palpation. The patient was referred to cone beam computed tomography (CBCT) for further evaluation of the left condyle. The visualization and reconstructions of the CBCT were performed using the RadiAnt Dicom Viewer (Medixant, Poland), version 5.5. The CBCT showed a left TMC (Figure 5), which is clearly observed on the Maximum Intensity Projection (MIP) reconstruction, where the 3 heads of the left mandibular condyle are seen simultaneously compared to the right normal condyle. The three-dimensional (3D) CBCT reconstruction allows us to evaluate the external shape of the left TMC compared to the normal right condyle (Figure 6) and to explore it in different planes (Figure 7). A diagnostic intra-ligamentary anesthesia was performed around tooth no. #48 resulting in complete elimination of the familiar pain in the right masseter leading to a diagnosis of referred pain due to pericoronitis of tooth no. #48, extraction of teeth no. #18, no. #48 was recommended. Concerning the non-familiar pain of left masseter, temporalis and right temporalis, the patient reported an awake bruxism behavior which was in accord with the intra-oral findings on the buccal mucosa and the tongue, and therefore, a diagnosis of Myalgia secondary to probable awake bruxism was made. Awareness of the behavior and muscle relaxing exercises were recommended. Finally, concerning the left TMC suspected to be a degenerative joint disease, due to the asymptomatic course of the condition, clinical and radiological follow-up after one year was recommended. According to the findings, a decision on the need for further follow-up will be made.

## 3. Material and Methods

A narrative review was performed using the following database engines: PubMed, EMBASE, Scopus, Web of science, Scientific Electronic Library Online, Cochrane and CINAHL, which were searched for TMC cases from inception until April of 2022.

Inclusion criteria were case reports of trifid mandibular condyle in the English language.

## 4. Results

Only 13 case reports of TMC were found in the literature [1,2,3,4,5,6,7,8,16,17,19,20,21]. Results of the narrative review are summarized in Table 1.

## 5. Discussion

As mentioned above, to date TMC is a rare anatomical variant of the mandibular condyle [1,2,3,4,5,6,7,8]. Only 13 previous cases were found in the literature.

The first case of TMC was reported by Artvinli and Kansu in the year 2003 [16], while the last one was reported by Gonz’alez-Garrido L. et al. in 2022 [22]. One unique tetrafid condyle case was reported by Sahman H. et al. in the year 2012 [13]. Definition criteria for bifid mandibular condyle has evolved since 1941, when Hrdlicka defined it as a condylar split or groove of variable depth.

The definition was changed in 1998 by Stefanou et al. describing it as the formation of two distinct condyles with a single separate neck irrespective of whether the heads are oriented mediolaterally or anteroposteriorly. In 2008, Dennison et al. claimed that “true BMC” must have an anterioposterior orientation. However, in 2010, Lopez-Lopez et al. opposed the statement of orientation and suggested that in order to call BMC a true BMC, the two heads must emerge from the condylar neck [2].

Bifid condyle may be classified by its orientation. Shriki J. et al. [23] and Szentpetery A. et al. [24] suggested that mediolateral split condyles are caused by a developmental anomaly, while anteroposterior oriented condyles are caused by trauma.

There are two main theories of possible etiology of BMC [16,25,26,27,28]. The first is trauma; this theory is based on the fact that after a fracture to the condylar neck, the force of lateral pterygoid muscle displaces the condyle in an anteromedial direction. Then, metaplasia of local fibroblasts in the condylar neck produce a new condylar head at the normal anatomical site while the original displaced condylar head undergoes resorption. As a result of this action, the traumatized joint will have two or more condyles for a while—the original displaced condyle which is considered a nonfunctioning condyle and the new formed condyle, which is the functioning one—until the original condyle is resorbed [5,15,29].

Experimental studies which were conducted on animals, demonstrated a high capacity of regeneration and remodeling of the condyle after trauma. For example, lizards sharing the same toolbox of genes as humans have the ability to form a new tail after an autotomy [2]. In fact, 10 out of 14 trifid mandibular condyle cases (13 published case reports plus this case) reported a history of trauma [2,3,4,5,7,8,16,19,20]. On the other hand, 4 out of 14 cases did not report history of trauma [1,6,17,21], which suggests the other possible etiology which is a developmental etiology. Hrdlicka assumed that blood flow obstruction during the development of the condyle may cause a division condylar cartilage at the age of 20 weeks in embryos. This septum is supposed to be resorbed and disappear by the age of 19 months of life. Injury or continuous shortage of blood supply may affect the ossification of the mandible and cause the splitting of the condyle into two heads [23,24]. Other causes such as endocrine disorders, nutritional deficiency, infection, trauma, irradiation and genetic factors were suggested [16]. Regarding symptoms of BMC, Loh F. and Yeo J. reported 67% of asymptomatic cases [30]. Cho b. and Chung y. reported 49% of asymptomatic cases in their retrospective study [31]. Fererres J. et al. reported a 40.6% of asymptomatic cases in their systematic review [15]. In symptomatic cases, hypomobility, arthralgia, articular noise and ankylosis were the most reported signs and symptoms [15]. Isik et al., reported a case of severe open bite and only mandibular first molar contact in a 6-year-old boy three months after a fall during running [32]. Some authors suggested that in many bifid condyle cases arthritic changes were demonstrated; in addition, it was suggested that ostreoarthrosis might develop in bifid condyles due to traumatic etiology [5]. These findings are in accord with the results of the review (Table 1) demonstrating that five of the described cases were totally asymptomatic and incidentally detected during a regular radiographic control of the patient, implementing that an underdiagnosis is highly probable [1,2,3,4,17]. Signs and symptoms that were reported in TMC cases include: pain, anterior open bite, deviation of the mandible, jaw weakness, mandibular hypoplasia, restricted mouth opening, snoring during sleep, noises, Frey’s syndrome, facial asymmetry and swelling of the masseteric region and incisal midline shift. Antoniades K. et al. reported a case of TMC of a 15-year-old boy who reported trauma at the age of 5 years old, with mandibular hypoplasia, restricted mouth opening and snoring during sleep, the skeletal relationship was bilaterally Angle Class II division 1 with traumatic occlusion [20]. These signs and symptoms suggest that TMC can potentially cause a variety of problems, such as temporomandibular disorders, occlusal problems and TMJ development. There is not enough information to conclude whether these signs and symptoms developed as a result of TMC, or they developed without relation to TMC. In addition, there are not enough studies on the long-term effect of TMC on the TMJ; the only existing information relies on case reports published in the literature. 

TMC is increasingly being detected as more advanced imaging techniques, especially CBCT, computed tomography (CT) and magnetic resonance imaging (MRI) are emerging in the field of dentistry [12,13,14,15]. 

Despite this, etiology and pathogenesis are still unclear [33,34,35,36].

According to the existing literature, the average age of diagnosis of TMC is 24 (6–52)—excluding the case of the skeletal individual which was discovered by Gonz’alez-Garrido L. et al. (Table 1).

The report of TMC is equally distributed between both sexes with 7 males and 7 females [1,2,3,4,5,6,7,8,16,17,19,20,21] (Table 1).

Treatment modalities that were suggested for BMC include analgesics and anti-inflammatory agents, muscle relaxants, physiotherapy and splints which are related generally to TMJ pain [5]. 

Regarding the treatment of TMC, 6 out of 14 cases did not suggest any treatment as they were asymptomatic [1,2,4,5,6,17], and 3 out of 14, including the present case report, recommended follow up [3,8,16]. In Antoniades K. et al.’s TMC case report of a 15-year-old boy presented with mandibular hypoplasia, restricted mouth opening and snoring during sleep, it was recommended that the boy use muscle relaxants, occlusal splint and a wooden tongue spatula as treatment; an improvement of mouth opening and the mandibular movements during function was also reported [20].

Motta-Junior J. et al. reported a case of a 17-year-old male with TMC who was diagnosed with Frey’s syndrome and was treated with subcutaneous injection of botulinum toxin A (BTA) with a follow up of 2 years with no complaints [21].

Jha A. et al. reported a case of a 6-year-old child with TMC and BMC with a history of trauma at the age of 4 who suffered severe restriction of movement of TMJ and was treated with physiotherapy and NSAIDs [7], while Çagirankaya and Hatipoglu reported a follow up of 1 year after prosthetic rehabilitation with no alteration in TMJ functions [17].

As a summary, the main points to be learned are that most cases of TMC reported a history of trauma in childhood, highlighting the importance of taking a thorough medical and dental history including questions about past trauma to the jaw and face, in addition to the importance of making a detailed extra- and intra-oral examination. In the present case, the facial asymmetry and the midline shift were red flags.

## 6. Conclusions

Trifid mandibular condyle, although being a rare anatomical entity, is increasingly being detected as more advanced imaging techniques, especially CT, CBCT and MRI, are emerging in the field of dentistry [37]. The etiology and pathogenesis of TMC, as well as possible developmental variants and the relationship with TMD are still unclear. Further studies and follow-up may help to better understand this anatomic variant and possible interactions with local pathologies.

## Figures and Tables

**Figure 1 life-12-00976-f001:**
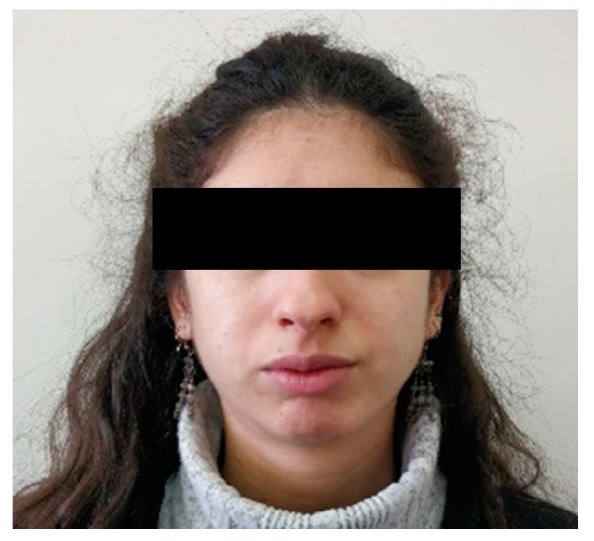
An extraoral photograph of the patient.

**Figure 2 life-12-00976-f002:**
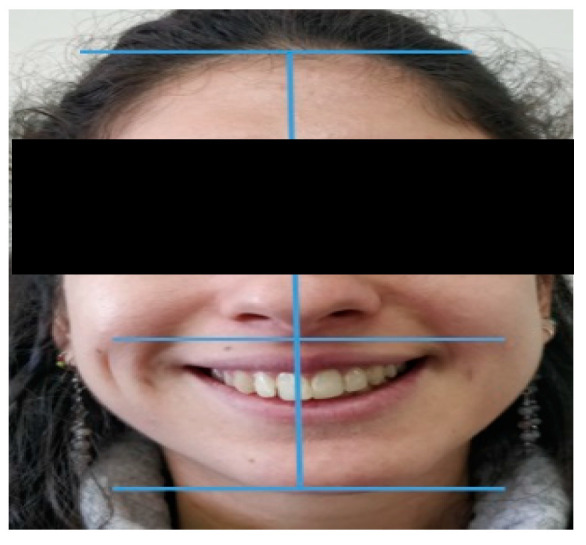
Note chin deviation to the left.

**Figure 3 life-12-00976-f003:**
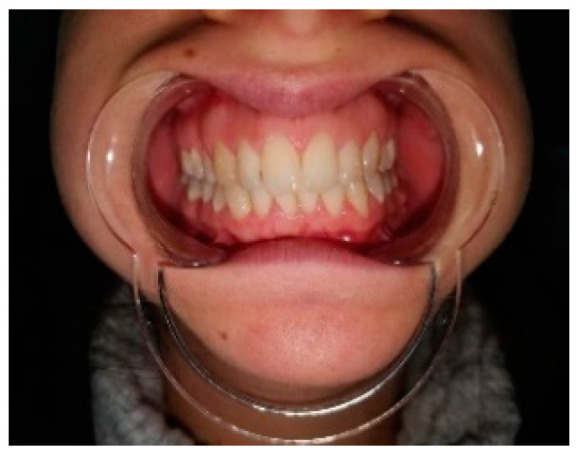
Note the 3 mm midline shift to the right.

**Figure 4 life-12-00976-f004:**
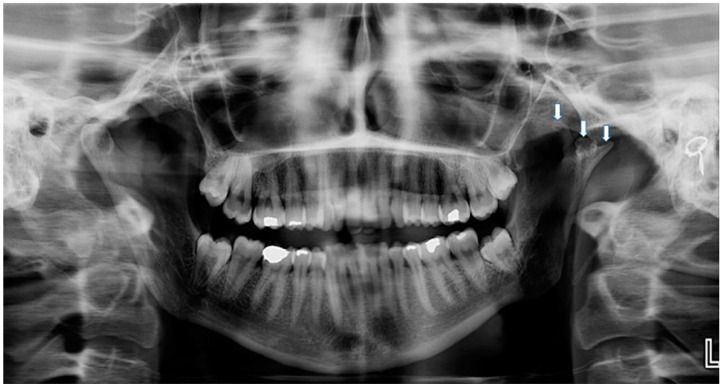
Panoramic X-ray demonstrating a partially impacted right and left lower wisdom tooth and a left bifid mandibular condyle with suspected lesion.

**Figure 5 life-12-00976-f005:**
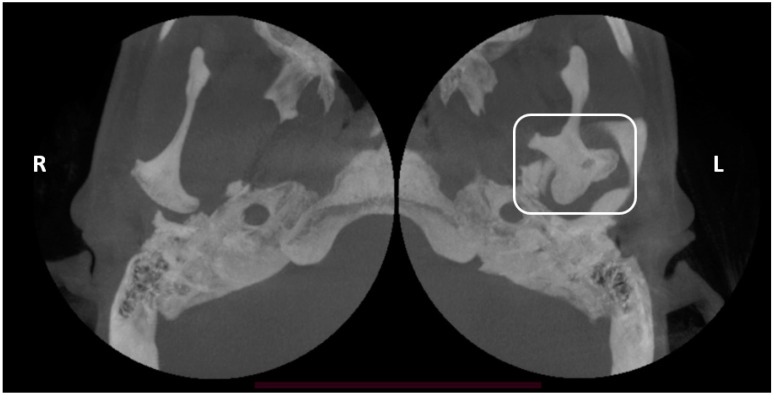
Maximum intensity projection (MIP) reconstruction of CBCT showing 3 heads of the left mandibular condyle (rectangle) compared to the right normal condyle.

**Figure 6 life-12-00976-f006:**
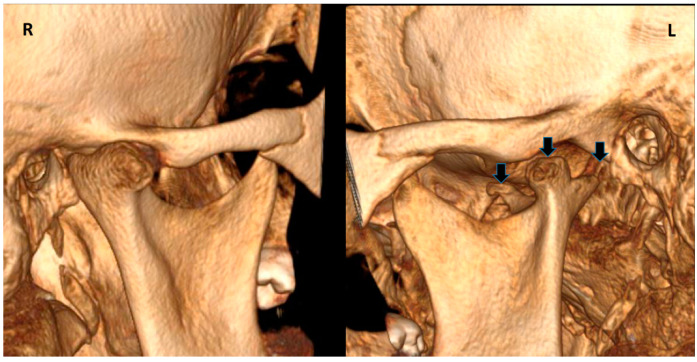
3D CBCT reconstruction showing 3 heads of the left mandibular condyle (arrows) compared to the normal structure of the right condyle.

**Figure 7 life-12-00976-f007:**
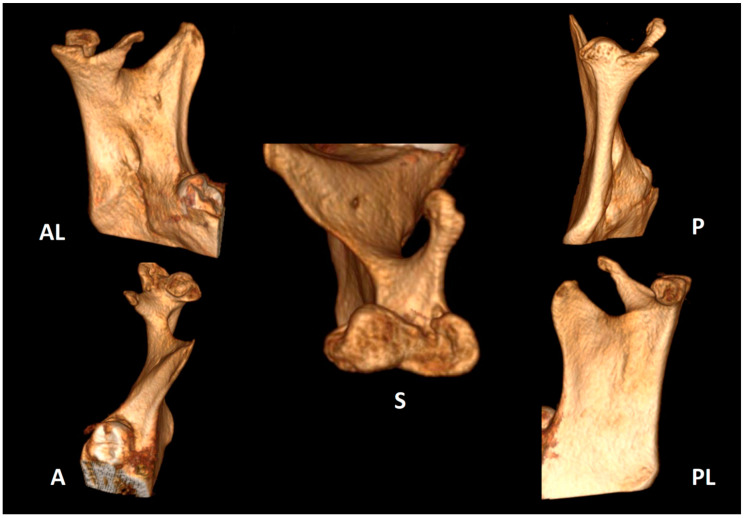
3D CBCT reconstruction in different planes (A—anterior; AL—anterior lateral; S—superior; P—posterior; PL—posterior lateral) showing the 3 heads of the left mandibular condyle.

**Table 1 life-12-00976-t001:** Summary of the narrative review results.

Author, Year	Gender, Age	Signs & Symptoms	Etiology	Treatment	Follow Up
Artvinli and Kansu 2003 [16]	F, 25	Anterior open bite Slight deviation of the mandible to the left Minor weakness after chewing	Trauma	FU	-
Antoniades K. et al., 2004 [20]	M, 15	Mandibular hypoplasia, restricted mouth opening Snoring during sleep	Trauma	Muscle relaxants, occlusal splint, and wooden tongue spatulas	-
Çagirankaya & Hatipoglu 2005 [17]	F, 52	Mandible deviated to the right	-	None	No alteration in TMJ functions one year after prosthetic rehabilitation
Sezgin & Katipman 2009 [3]	M, 31	Mandible deviated to the right	Trauma	FU	-
Rodrigo Millas M. et al., 2010 [6]	F, 27	Noise, click and unilateral TMJ pain Progressive joint hypomobility	-	-	-
Motta-Junior J. et al., 2012 [21]	M, 17	Frey’s syndrome	-	Subcutaneous injection of botulinum toxin A (BTA)	2 years with no complains
Jha A. et al., 2013 [7]	M, 6	Severe restriction of movements of TMJ	Trauma	Physiotherapy & NSAIDs	-
Warhekar A. et al., 2014 [4]	F, 37	Bilaterally asymmetrical face Diffuse painless swelling over the right masseteric region	Trauma	None	-
Prasanna T. et al., 2015 [1]	F, 26	Mild facial asymmetry, micrognathia & deviation of the mandible to left	-	None	-
Hernández-Andara A. et al., 2017 [8]	M, 12	Facial asymmetry & a clicking noise in the left TMJ	Trauma	FU	-
Ayat A. et al., 2018 [5]	F, 40	Chin deviation to the right Pain in the right mandibular molar area	Trauma	None	-
Orhan Güven 2018 [2]	M, 19	Chin deviation	Trauma	None	-
González-Garrido L. et al., 2022 [19]	M, +40 (skeletal individual)	-	Trauma	-	-

(M—male; F—female; FU—follow up; (-)—not mentioned).

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
