# Peer review of "Trifid Mandibular Condyle: Case Report and Current Review of the Literature"

_life, 2022, doi:10.3390/life12070976_

Round 1

Reviewer 1 Report

This manuscript reported a new case of trifid mandibular condyle (TMC) and summarizing other cases by literature review. Trifid mandibular condyle is a rare morphological abnormality of mandibular condyle which may be caused by either developmental defect or trauma. This report is useful to understand TMC phenotype. However, the authors did not mention about the method of literature review and ignored the manuscript style. Thus, there are several points which are needed to improve.

1.     Figure 6. Please provide normal right-side images as references. Image of “P” is not posterior; this image is tilted lateral posterior. Please provide correct image.

2.     Subsection “Discussion” is the results of literature review. All results should be move to subsection “Results”.

3.     There is no detailed information about literature review (systematic review); for example, inclusion/exclusion criteria and period (year) when the manuscripts are searched.

4.     Please discuss about TMC from several point of view in the discussion such as how TMC affect to TMJ development, occlusion, intra-articular trauma/fracture, associated with temporomandibular disorder, etc.

Author Response

Dear Reviewer,

First, we would like to thank you for taking the time to review our case report and review, and especially for your valuable comments and thorough review and comments which helped improve the manuscript.

Below please find our point-by-point response.

  1. Figure 6. Please provide normal right-side images as references. Image of “P” is not posterior; this image is tilted lateral posterior. Please provide correct image.

Response: Done.

  1. Subsection “Discussion” is the results of literature review. All results should be move to subsection “Results”.:

Response: Done.

  1. There is no detailed information about literature review (systematic review); for example, inclusion/exclusion criteria and period (year) when the manuscripts are searched.

Response: Added to Material and method section.

  1. Please discuss about TMC from several point of view in the discussion such as how TMC affect to TMJ development, occlusion, intra-articular trauma/fracture, associated with temporomandibular disorder, etc.

Response: Added to section discussion.

Reviewer 2 Report

Dear authors,

Thanks for submitting your manuscript entitled "Trifid mandibular condyle: a case report and current review of the literature".

I have carefully reviewed your report and I would like to address the following:

1. Introduction is too short, because it is only one paragraph. Please extend the information regarding the difference between TMC and BMC. Describe more in details the variety of symptoms.

2. In you case report: patient is 26 years old and pain appeared 3 years before. I would like to know if patient has never had a radiograph showing the before in her 26 years of age or probably she has had dental care but no diagnosed. I mention this because a panoramic radiograph is a standard image for a regular evaluation in dentistry. Provide more information about dental history.

3. Please provide extra-oral evaluation of the patient. Mention if you see any facial asymmetry.

4. Your evaluation sequence is not written in an ideal way. Dental history includes questions of any trauma in face, chin or teeth in the past, and dental history is before intra-oral evaluation. However, you asked patient about trauma after evaluating the radiograph (after intra-oral evaluation).

5. Thanks for the great CBCT images.

6. In your discussion section, you are describing many case reports, but please also include if there were follow-up of those cases.

Thanks.

Author Response

Dear Reviewer,

First, we would like to thank you for taking the time to review our case report and review, and especially for your valuable comments and thorough review and comments which helped improve the manuscript.

Below please find our point-by-point response.

  1. Introduction is too short, because it is only one paragraph. Please extend the information regarding the difference between TMC and BMC. Describe more in details the variety of symptoms.

 Response: Done, added more information as requested in Introduction section.

  1. In you case report: patient is 26 years old and pain appeared 3 years before. I would like to know if patient has never had a radiograph showing the before in her 26 years of age or probably, she has had dental care but no diagnosed. I mention this because a panoramic radiograph is a standard image for a regular evaluation in dentistry. Provide more information about dental history.

Response: Thank you for your comment, you are absolutely right. A previous panoramic XR should have been a part of dental history of every patient at the age of 26. Unfortunately, this patient is a part of the t underprivileged patient clinic at the dental school. All her treatments were provided by the school. Therfore it wad her first panoramic xr.

  1. Please provide extra-oral evaluation of the patient. Mention if you see any facial asymmetry.

Response: Added in the text, and figures.

  1. Your evaluation sequence is not written in an ideal way. Dental history includes questions of any trauma in face, chin or teeth in the past, and dental history is before intra-oral evaluation. However, you asked patient about trauma after evaluating the radiograph (after intra-oral evaluation).

Response: Thank you for pointing out this issue. The sequence was corrected in the text.

  1. Thanks for the great CBCT images. ?.
  2. In your discussion section, you are describing many case reports, but please also include if there were follow-up of those cases.

Response: Added in the discussion section and also a column was added to table.1